# Workplace noise exposure and the prevalence and 10-year incidence of age-related hearing loss

Bamini Gopinath[1,2]*, Catherine McMahon[1], Diana Tang[1,2], George Burlutsky[2], Paul Mitchell[2]

1 Department of Linguistics, Macquarie University, Sydney, NSW, Australia, 2 Centre for Vision Research, Department of Ophthalmology and Westmead Institute for Medical Research, University of Sydney, NSW, Australia

* bamini.gopinath@mq.edu.au

**Data Availability Statement:** All relevant data are within the paper.

**Funding:** PM received an Australian National Health and Medical Research Council (Grant Nos.

## Abstract

There is paucity of population-based data on occupational noise exposure and risk of age-related hearing loss. Therefore, we assessed cross-sectional and longitudinal associations of past workplace noise exposure with hearing loss in older adults. At baseline, 1923 participants aged 50+ years with audiological and occupational noise exposure data included for analysis. The pure-tone average of frequencies 0.5, 1.0, 2.0 and 4.0 kHz ($PTA_{0.5-4KHz}$) >25 dB HL in the better ear, established the presence of hearing loss. Participants reported exposure to workplace noise, and the severity and duration of this exposure. Prior occupational noise exposure was associated with a 2-fold increased odds of moderate-to-severe hearing loss: multivariable-adjusted OR 2.35 (95% CI 1.45–3.79). Exposure to workplace noise for >10 years increased the odds of having any hearing loss (OR 2.39, 95% CI 1.37–4.19) and moderate-to-severe hearing loss (OR 6.80, 95% CI 2.97–15.60). Among participants reporting past workplace noise exposure at baseline the 10-year incidence of hearing loss was 35.5% versus 29.1% in those who had no workplace noise exposure. Workplace noise exposure was associated with a greater risk of incident hearing loss during the 10-year follow-up: multivariable-adjusted OR 1.39 (95% CI 1.13–1.71). Prior occupational noise exposure was not associated with hearing loss progression. Workplace noise exposure increased the risk of incident hearing loss in older adults. Our findings underscore the importance of preventive measures which diminish noise exposure in the workplace, which could potentially contribute towards reducing the burden of hearing loss in later life.

## Introduction

Age-related hearing loss is the most frequent communication disorder and is associated with a greater risk of depression, impairs quality of life and the ability to conduct activities of daily living [1–5]. It is a multifactorial process, resulting primarily from the accumulating effects of noise and ageing on the cochlea [6]. Ageing-related degeneration may negatively impact

974159, 991407, 211069, 262120). The funders had no role in study design, data collection and analysis, decision to publish, or preparation of the manuscript.

**Competing interests:** The authors have declared that no competing interests exist.

cochlear hair cells and the vascular supply to the cochlea, and the transmission of auditory information along the neural pathway, leading to impaired hearing [5, 6]. Chronic noise exposure is also thought to be responsible for both mechanical and metabolic damage to the cochlea [7], especially the cochlear hair cells, and through hypoxia caused by noise-induced capillary vasoconstriction [7, 8].

Animal models of auditory ageing have shown that repeated short-duration loud sound overstimulation accelerates the time-course of age-related hearing loss [9]. There is a dearth of longitudinal population-based studies that have investigated noise exposure and hearing decline, and observational studies have thus far, provided equivocal findings on the relationship between noise exposure and subsequent hearing loss in adults. The Framingham Heart Study [10] showed that in men, reporting noise exposure subsequent hearing loss progression in later age was exacerbated even at frequencies outside the range of the original noise-induced hearing loss. Conversely, other prospective cohort studies such as the Epidemiology of Hearing Loss Study [11] and a Swedish study of older adults [12] were not able to show an independent association between occupational noise exposure and long-term hearing function.

Age-related and noise-induced hearing loss have important public health implications [5, 7, 10], given the high prevalence of both occupational noise exposure and age-related hearing loss [4, 7]. Despite this substantial burden, there are very few large population-based studies that have evaluated both the cross-sectional and longitudinal contribution of noise exposure to age-related hearing loss. Therefore, in this study we aimed to explore the prevalence, 10-year incidence and progression of hearing loss associated with occupational noise exposure among older adults.

## Materials and methods

The Blue Mountains Hearing Study (BMHS) was a population-based survey of age-related hearing loss conducted during 1997–2007 among participants of the Blue Mountains Eye Study (BMES) cohort [13]. During 1992–4, 3,654 participants aged 49 years or older were examined (82.4% participation; BMES-1). Surviving baseline participants were invited to attend 5-year follow-up examinations (1997–9, BMES-2), at which 2334 (75.1% of survivors) and an additional 1174 newly eligible residents were examined, i.e. those who had moved into the study area or study age group (Extension Study). At the 10-year follow-up (2002–4, BMES-3) and 15-year follow-up (2007–9, BMES-4), 1952 participants (75.6% of BMES-1 survivors) and 1149 (55.4% of s BMES-1 survivors) were re-examined, respectively. Hearing was measured from BMES-2 (i.e. 1997–99) onwards i.e. 2956 participants aged $\geq$50 y had audiometric testing performed at BMES-2 (i.e. BMHS). The study was approved by the Human Research Ethics Committee of the University of Sydney and was conducted adhering to the tenets of the Helsinki Declaration. Signed informed consent was obtained from all the participants at each examination.

### Audiological examination

Pure-tone audiometry at both visits was performed by audiologists in sound-treated booths, using standard TDH-39 earphones and Madsen OB822 audiometers (Madsen Electronics, Copenhagen, Denmark), calibrated regularly to Australian standards. Audiometric thresholds for air-conduction stimuli in both ears were established for frequencies at 250, 500, 1000, 2000, 4000, 6000 and 8000 Hz. We determined hearing impairment as the pure-tone average of audiometric hearing thresholds at 500,1000, 2000, and 4000 Hz ($PTA_{0.5-4KHz}$), defining any level of hearing loss as $PTA_{0.5-4KHz} > 25$ dB hearing level (HL), in the better of the two ears. This defined hearing loss as bilateral. Low frequency hearing loss was defined as a $PTA_{0.5, 1, 2 KHz} > 25$ dB HL

in the better of the two ears. High frequency hearing loss was defined as a $PTA_{4, 6, 8 KHz} > 40$ dB HL in the better of the two ears. These classifications are the same as in The Epidemiology of Hearing Loss Study [14]. Bone conduction was also evaluated whenever air conduction thresholds were greater than 15-dB hearing level (dB HL) at 4 frequencies (500, 1000, 2000, and 4000 Hz). Participants were examined for any evidence of collapsed canals, and if present, air conduction thresholds at the higher frequencies were reassessed, taking care to reduce pressure on the external ear. The audiologist also performed otoscopic evaluation and examined the ears for wax occlusion, and if present, the participants were asked to return for assessment after treatment.

A person was considered at risk for incident bilateral hearing loss during the 10-year period (from BMES-2 to BMES-4) if the $PTA_{0.5-4KHz}$ in the better ear was $\leq 25$ dB HL at BMES-2 or BMES-3. Incident bilateral hearing loss was defined as a $PTA_{0.5-4KHz} > 25$ dB HL in the better ear at the 5- or 10-year follow-up examination among participants without hearing loss.

### Assessment of study factor (occupational noise exposure) and potential covariates

A comprehensive medical history that included information about hearing, demographic factors, socio-economic characteristics and lifestyle factors, was obtained from study participants. The medical history included presence of cardiovascular or other systemic disease and associated risk factors, medications used, exercise, smoking, and consumption of caffeine and alcohol.

An audiologist also asked questions around history of any self-perceived hearing problem, including its severity, onset and duration, whether help was sought for this from primary care practitioners or other professionals, and if a hearing aid was provided. Additional questions included family history of hearing loss, past medical and/or surgical treatment of otologic conditions, other diseases associated with hearing loss, and risk factors for impaired hearing. Exposure to noise at work, or during military service or leisure activities was determined by asking the following: 'Have you ever worked in a noisy industry or noisy farm environment?' Based on whether a respondent answered 'yes' or 'no' to this question, they were classified as having any exposure to occupational noise or no exposure to occupational noise, respectively, and this was the specific variable included in statistical models. Participants were also asked for how long a period did he or she worked in this industry: <1 year; 1 to 5 years; 6 to 10 years; or more than 10 years? Subjects were also asked to described the noise level that they were exposed to on an average day with the following options: mostly quiet; tolerable but able to hear speech; or unable to hear anyone speaking. Participants were classified as being exposed to 'severe' noise levels, if their response was that they were 'unable to hear anyone speak', all other responses were classified as a 'tolerable' level of noise. They were asked whether they usually did or did not wear hearing protection. Details were obtained on whether they had done any of the following work or activities: band music; woodwork; carpentry; sheet metal work; chain sawing; used power tools; driven or worked on racing cars; used personal electronic music players; or attended loud concerts or band performances. They were also asked whether they had been exposed to the noise of gunfire or explosions.

### Statistical analysis

SAS statistical software (SAS Institute, Cary NC) version 9.4 was used for analysis including t-tests, $\chi^2$-tests and logistic regression. The association between occupational noise exposure and prevalence of hearing loss was examined in logistic regression models, adjusting first for age and sex, and then further adjusting for confounders previously found to be significantly

associated with hearing loss prevalence (education, smoking, previous history of diagnosed stroke and diabetes, and family history of hearing loss). Results of this cross-sectional analyses are expressed as adjusted odds ratios (OR) with 95% confidence intervals (CI). For 10-year incidence of age-related hearing loss (study outcome), risk ratios (RR) were obtained by using linear fixed effects for longitudinal data i.e. using Poisson Regression approach to prospective studies with Log as link function. The multivariable model for this incidence analyses involved adjustments for age, sex and family history of hearing loss. Analysis of covariance (least-squares means) was used to obtain adjusted mean hearing thresholds. $P$-values <0.05 indicated statistical significance.

## Results

Among participants with complete audiological data at baseline (n = 2015), 68 participants were excluded on the basis that they had conductive hearing loss, middle ear hearing loss, childhood hearing loss and/or a history of diagnosed otosclerosis. An additional15 participants were excluded as they reported previous head injury. This resulted in a total of 1932 subjects with complete audiological and occupational noise exposure data, including 679 participants who reported previous exposure to occupational noise and 1244 with no prior exposure. Persons reporting occupational noise exposure were approximately one year younger (69.2 years) than those without prior exposure (70.8 years; p = 0.004) and they were also more likely to be: male (70.0% versus 27.7%, p <0.001), a smoker (11.4% versus 7.3%, p = 0.02), without tertiary qualifications (43.0% versus 37.2%, p = 0.014) and with type 2 diabetes (14.1% versus 10.0%, p = 0.009). However, participants with prior noise exposure were less likely to have a family history of hearing loss (38.7% versus 44.7%, p = 0.011). Fig 1 shows the type of occupational or recreational noise that BMHS participants were exposed to, with power tools (66.8%) and gun noise (56.2%) the most frequent types of noise exposure reported. Of those exposed to occupational noise, only 68 (10.0%) used any type of hearing protection device.

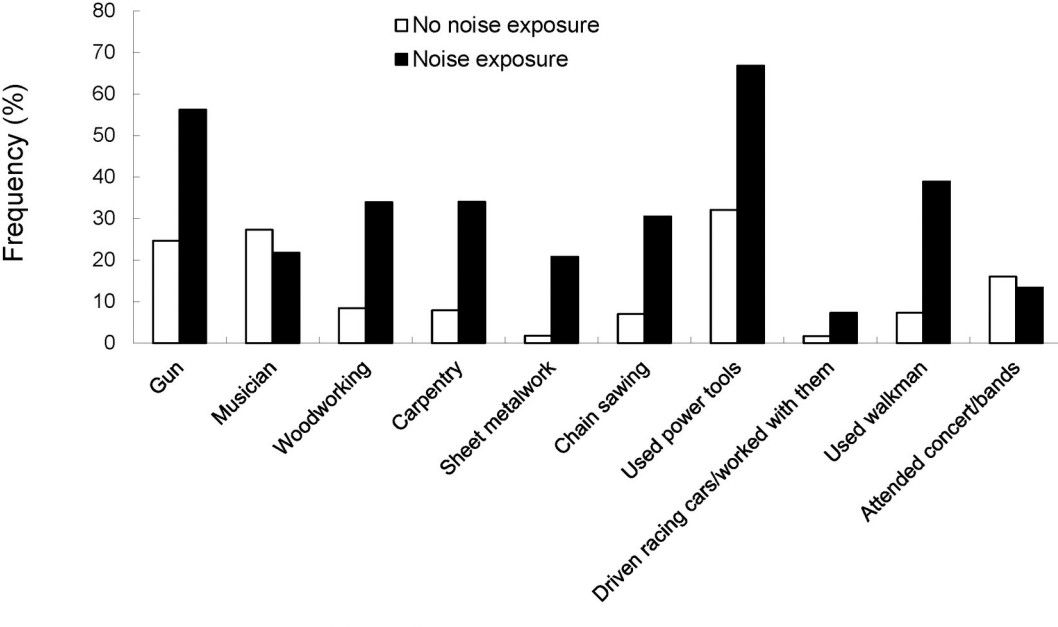

**Fig 1. Type of occupational or recreational noise exposure reported by participants of the Blue Mountains Hearing Study at baseline (1997–9).**

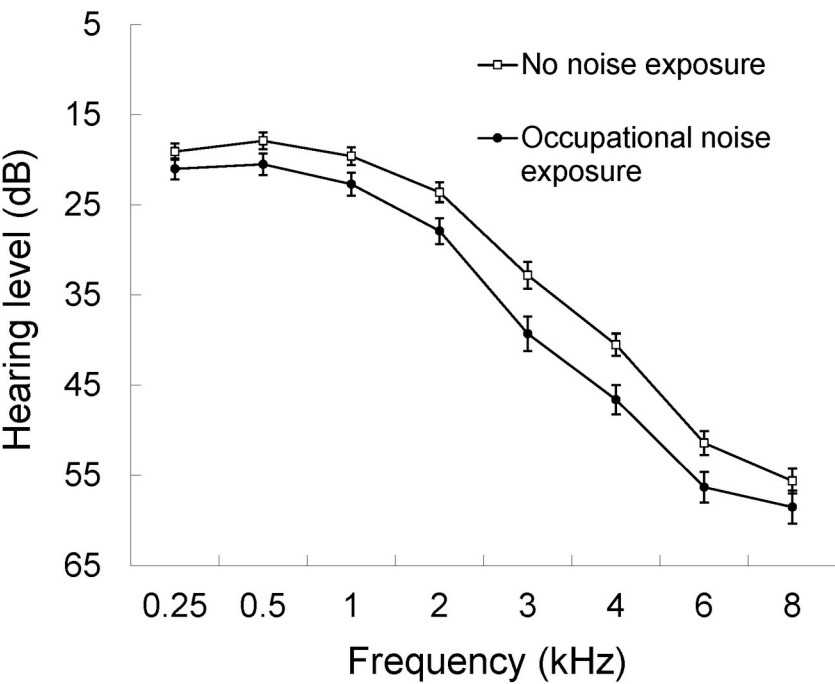

**Fig 2. Mean air-conduction thresholds at baseline in those with or without occupational noise exposure.** Age- and sex-adjusted mean hearing thresholds for participants who did not report occupational noise exposure, pure-tone average threshold 0.25–8 kHz, ≤ 25 dB HL (□); age and sex adjusted mean hearing thresholds for participants who reported occupational noise exposure, pure-tone average threshold 0.25–8 kHz, > 25 dB HL (•).

Age-sex adjusted pure-tone air-conduction audiometric thresholds are shown in Fig 2 for frequencies between 0.5–8.0 kHz in participants exposed and not exposed to occupational noise. Hearing sensitivity was worse (higher thresholds) for all 8 frequencies in subjects exposed to occupational noise, with statistically significant differences at each frequency. Prevalence of any hearing impairment ($PTA_{0.5-4KHz}$ >25 dB HL in the better ear) was 44.9% in subjects with prior occupational noise exposure, compared to 36.4% in those not exposed. Participants reporting exposure to noise in the workplace had a 56% higher likelihood of having hearing loss, OR 1.56 (95% CI 1.21–2.02), after adjusting for age, sex, qualification, smoking, stroke, type 2 diabetes and a family history of hearing loss. This likelihood increased to 2-fold when the outcome was prevalence of moderate to severe hearing loss, OR 2.35 (95% CI 1.45–3.79).

Being exposed to occupational noise for >10 years was significantly associated with mild (OR 1.62, 95% CI 1.16–2.25) and moderate-to-severe hearing loss (OR 3.61, 95% CI 2.02–6.43), after multivariable adjustment (Table 1). Similarly, severe occupational noise exposure was associated with a 2- and 3-fold increased likelihood of mild and moderate to severe hearing impairment, respectively (Table 1). Table 2 shows the three combinations of severity and duration of occupational noise exposure. Compared to those without exposure to occupation noise, participants reporting greater than 10 years of severe occupational noise exposure had the highest odds of having any level of hearing loss (OR 2.39, 95% CI 1.37–4.19).

Of the 1923 participants examined at baseline, 895 participants who did not have hearing loss at baseline and who had their hearing assessed at the 5- and/or 10-year follow-up were included in temporal analysis. The 10-year incidence of hearing loss was 35.5% in participants exposed to occupational noise compared to 29.1% of subjects without noise exposure (Table 3). Participants reporting any past noise exposure at baseline had 39% higher risk of

**Table 1. Association between severity and duration of exposure to workplace noise and prevalent hearing loss, presented as Odds Ratio (OR) and 95% Confidence Interval (CI).**

| Noise exposure | Any loss (>25 dB HL) | | | Mild loss (>25 - ≤40 dB HL) | | | Moderate-severe loss (>40 dB HL) | | |
|---|---|---|---|---|---|---|---|---|---|
| | N (%) | Age-sex adjusted OR (95% CI) | Multivariable adjusted OR (95% CI) [a] | N (%) | Age-sex adjusted OR (95% CI) | Multivariable adjusted OR (95% CI) [a] | N (%) | Age-sex adjusted OR (95% CI) | Multivariable adjusted OR (95% CI) [a] |
| Duration | | | | | | | | | |
| 1–10 yrs | 129 (43) | 1.47 (1.09–1.98) | 1.37 (1.00–1.88) | 105 (35) | 1.53 (1.12–2.09) | 1.39 (1.00–1.94) | 24 (8) | 1.65 (0.95–2.86) | 1.59 (0.85–2.94) |
| >10 yrs | 175 (47) | 1.80 (1.35–2.42) | 1.79 (1.31–2.45) | 127 (34) | 1.68 (1.23–2.30) | 1.62 (1.16–2.25) | 48 (13) | 2.94 (1.77–4.90) | 3.61 (2.02–6.43) |
| Severity | | | | | | | | | |
| Tolerable | 209 (42) | 1.45 (1.12–1.88) | 1.40 (1.07–1.85) | 160 (32) | 1.42 (1.08–1.87) | 1.35 (1.01–1.80) | 49 (10) | 2.08 (1.32–3.30) | 2.14 (1.27–3.60) |
| Severe | 84 (54) | 2.23 (1.51–3.29) | 2.16 (1.43–3.27) | 63 (40) | 2.15 (1.43–3.24) | 2.00 (1.29–3.08) | 21 (13) | 2.78 (1.44–5.37) | 3.41 (1.68–6.93) |

[a] Additional adjustment for age, sex, qualification, smoking, stroke, type 2 diabetes and a family history of hearing loss.

developing hearing loss at 10-year follow-up multivariable-adjusted OR 1.39 (95% CI 1.13–1.71). Compared to participants who were not exposed to workplace noise at baseline, those participants who reported occupational noise exposure for a duration of 1–10 years and 10 years had a 40% and 44% increased risk of incident hearing loss, respectively, after adjusting for all potential confounders (Table 3). Progression or worsening of hearing impairment was not associated with past exposure to workplace noise reported at baseline, OR 1.10 (95% CI 0.85–1.42) after multivariable adjustment.

## Discussion

Prevention of exposure to workplace noise could be a potential modifiable risk factor for age-related hearing loss. We provide robust epidemiological data showing that nearly one in two older adults exposed to occupational noise experienced impaired hearing at baseline. Exposure to workplace noise was a significant, independent predictor of incident sensorineural hearing loss. However, exposure to noise in the workplace was not a significant risk factor for hearing lossprogression in older adults.

Hearing loss was prevalent in 44.9% of study participants reporting exposure to noise in the workplace at baseline. This is relatively higher than the 38% reported by *Ferrite et al.* [15] for those reporting occupational noise exposure, this could be due to the differing age range i.e.

**Table 2. Association between combined severity and duration of exposure to workplace noise and prevalent hearing loss, presented as Odds Ratio (OR) and 95% Confidence Interval (CI).**

| Noise exposure | Any loss (>25 dB HL) | | | Mild loss (>25 - ≤40 dB HL) | | | Moderate-severe loss (>40 dB HL) | | |
|---|---|---|---|---|---|---|---|---|---|
| | N (%) | Age-sex adjusted OR (95% CI) | Multivariable adjusted OR (95% CI) [a] | N (%) | Age-sex adjusted OR (95% CI) | Multivariable adjusted OR (95% CI) [a] | N (%) | Age-sex adjusted OR (95% CI) | Multivariable adjusted OR (95% CI) [a] |
| Not severe/ ≥10yrs | 127 (44) | 1.81 (1.09–2.99) | 1.90 (1.10–3.29) | 98 (34) | 2.10 (1.26–3.49) | 2.16 (1.24–3.75) | 29 (10) | 0.73 (0.23–2.33) | 0.70 (0.19–2.54) |
| Severe/ <10 yrs | 41 (50) | 1.54 (1.13–2.09) | 1.60 (1.15–2.22) | 37 (45) | 1.51 (1.09–2.10) | 1.55 (1.10–2.18) | 4 (5) | 1.99 (1.14–3.45) | 2.29 (1.22–4.28) |
| Severe/ ≥10yrs | 43 (58) | 2.62 (1.52–4.52) | 2.39 (1.37–4.19) | 26 (35) | 2.05 (1.13–3.75) | 1.77 (0.95–3.30) | 17 (23) | 5.36 (2.41–12.0) | 6.80 (2.97–15.6) |

[a] Additional adjustment for age, sex, qualification, smoking, stroke, type 2 diabetes and a family history of hearing loss.

**Table 3. Association between exposure to workplace noise and the 10-year incidence and progression of hearing loss, presented as Risk Ratios (RR) and 95% Confidence Interval (CI).**

| Exposure to workplace noise | Incidence of hearing loss [b] | | |
|---|---|---|---|
| | N (%) | Age-sex adjusted RR (95% CI) | Multivariable adjusted RR (95% CI) [a] |
| Any exposure | 114 (35.5) | 1.37 (1.11–1.68) | 1.39(1.13–1.71) |
| Duration | | | |
| 1–10 years | 53 (36.1) | 1.38 (1.08–1.76) | 1.40 (1.09–1.80) |
| >10 years | 60 (34.9) | 1.35 (1.04–1.75) | 1.44 (1.09–1.91) |

[a] 10-year incidence of bilateral hearing loss (pure-tone average of thresholds for 500, 1000, 2000 and 4000 Hz >25 dB HL in the better ear).

[b] Additional adjustment for family history of hearing loss.

the study sample was younger ranging from 41–55 years. The likelihood of having a hearing impairment at baseline increased with increasing severity and duration of noise exposure (either alone or in combination). This confirms the increasing prevalence of hearing loss with increasing duration of occupational noise exposure reported in a UK study of participants aged 16–64+ years [16]. Although, the odds of having moderate to severe hearing loss in our cohort was largely dependent on the severity of the noise exposure rather than the duration spent working in a noisy environment. Hence, while continuous noise exposure over the years is damaging, short exposures to high levels of noise in the workplace may be a more important contributor to impaired hearing late in life. Assessment of the potential health effects of such discontinuous noise exposure are limited and further research into this area is warranted.

Ours is one of the few cohort studies to show that exposure to workplace noise is a significant and independent predictor of incident hearing loss in older adults. Over one in three older adults reporting exposure to workplace noise developed incident hearing loss 10 years later. Moreover, increasing duration of occupational noise exposure at baseline was associated with increased risk of developing hearing loss over the 10-year follow-up. This observed association was independent of other hearing loss risk factors such as age and family history. These epidemiological data indicate that occupational noise exposure is likely to initiate the deterioration of the cochlear structures and that the ageing process could additionally contribute to this damage [10, 17]. Hence, noise and ageing could operate through common causal pathways, supporting the hypothesis that these factors interact in a biological additive model to result in hearing loss [7, 15]. The underlying pathways explaining this association could include hypoxia induced in the cochlea due to noise [15] and the degenerative changes with ageing which may affect neural fibres and parts of the cochlea [7].

We did not observe significant progression or worsening of hearing function due to prior exposure to workplace noise. This suggests that prior noise exposure does not damage the cochlear in a manner that continuously deteriorates hearing function over time [17, 18]. These findings support the study by *Albera et al.* [17] which showed that the progression of sensorineural hearing loss in individuals with noise-induced hearing loss was observed to be less than predicted in non-noise exposed individuals. The authors hypothesised that this was because in noise-exposed subjects, cochlear hair cells damaged by noise exposure are unlikely to be further damaged by ageing [17, 18].

We previously demonstrated in the BMHS [19] that the population attributable risk for occupational noise exposure was 20% and thus, it contributes substantially to the burden of age-related hearing loss, second only to a family history of hearing loss (22%). This attributable fraction is in agreement with data from the US National Institute for Occupational Safety and Health, showing that occupational noise is an important risk factor for hearing loss in workers

at most ages, contributing about 7 to 21% (averaging 16%) to the burden of adult-onset hearing loss globally [20]. However, we need to highlight that the contribution of occupational noise to hearing loss is likely to be complicated by the possible exposure of people to excessive noise in non-work settings and that other factors such as exposure to ototoxic substances or a number of medical conditions (e.g. diabetes), in addition to simply ageing, could contribute to the development of hearing loss.

In some occupations where hearing conservation methods are important and required, there is evidence of continuing poor compliance and limited audiometric screening [16]. It is known that compliance with wearing noise protection is not high [21] and that this appears to be related in part to the difficulties imposed by such protection upon communication with other workers, especially in an emergency [22]. Indeed, in our study cohort we observed only 10% of those reporting occupational noise exposure indicated using any form of hearing protection devices.

Based on our findings, a currently feasible approach to prevention is the timely recognition of noisy workplaces and a strict implementation of hearing conservation programs in these work environments. Currently, there is very low-quality evidence that the use of hearing protection devices in well-implemented hearing loss prevention programmes is linked to reduced hearing loss but this could not be demonstrated for other aspects, such as monitoring of noise levels, worker training or audiometry alone [23]. Additionally, engineering solutions such as new equipment, segregation of noisy equipment, installation of panels or curtains can significantly reduce noise levels as shown by case-control studies [23]. However, longitudinal research on the effects of engineering interventions to reduce noise is needed. For instance, field case studies with valid measures of personal noise doses of workers with long-term follow-up would provide better evidence than what is currently available [23]. At a minimum, for an effective and successful hearing conservation program—noise surveys and monitoring, employee education, training, and motivation, hearing protection equipment, audiometric testing, and record-keeping, as well as noise control are important [24]. Primary care physicians should aim to enquire about patient's noise exposure and to refer patients for elementary hearing conservation services (audiometry, counselling, and personal hearing protective devices) [25]. Moreover, audiologists and otolaryngologists see patients who have significant unprotected occupational and non-occupational noise exposure, and as such they play an important role in providing counselling, hearing protection and periodic audiometry for these patients [26]. Finally, clinical interventions such as the use of magnesium or antioxidants such as N-acetylecysteine for preventing noise-induced hearing loss may also hold some promise in the treatment of age-related hearing loss [7, 27].

Strengths of this study include relatively high participation rates, it longitudinal design and standardised, audiometric testing to measure hearing sensitivity rather than self-report, unlike many other studies that have investigated hearing loss and noise [16, 28]. Limitations of the study include potential systematic recall bias: participants self-reported duration and severity of noise exposure and participants who know they have a hearing impairment may more often recollect, or possibly even falsely recall exposure to noise compared to people with normal hearing. Such a bias may inflate the risk estimates.

## Conclusion

In conclusion, one in two participants exposed to occupational noise had some form of hearing loss at baseline and over one in three participants reporting baseline occupational noise exposure developed incident hearing loss 10 years later. These findings add to the evidence-base that age and occupational noise have a potential multiplicative effect on hearing function

and could act through common pathogenic pathways. This study highlights the potential burden due workplace noise exposure and the importance of public health policies that implement evidence-based interventions targeting exposure to occupational noise, which is likely to lead to a reduction in the prevalence and incidence of hearing impairment.

## Author Contributions

**Conceptualization:** Bamini Gopinath, Paul Mitchell.

**Data curation:** George Burlutsky.

**Formal analysis:** George Burlutsky.

**Funding acquisition:** Paul Mitchell.

**Investigation:** Bamini Gopinath, Catherine McMahon, Paul Mitchell.

**Methodology:** Bamini Gopinath, Catherine McMahon, Paul Mitchell.

**Project administration:** Paul Mitchell.

**Resources:** Paul Mitchell.

**Supervision:** Bamini Gopinath, Paul Mitchell.

**Writing – original draft:** Bamini Gopinath.

**Writing – review & editing:** Bamini Gopinath, Catherine McMahon, Diana Tang, George Burlutsky, Paul Mitchell.

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
