## [Decision Letter · Decision Letter 0]

6 May 2021

PONE-D-21-00280

Workplace Noise Exposure and the Prevalence and 10-Year Incidence of Age-Related Hearing Loss

PLOS ONE

Dear Dr. Gopinath,

Thank you for submitting your manuscript to PLOS ONE. After careful consideration, we feel that it has merit but does not fully meet PLOS ONE’s publication criteria as it currently stands. Therefore, we invite you to submit a revised version of the manuscript that addresses the points raised during the review process.

We apologise for the length of time to get the reviews to you.  The paper has now been reviewed by two reviewers who have raised a number of points in the manuscript needing attention or clarification.  A key point needing attention relates to the statistical analysis of prevalence and associations, along with more detail on the approach and choice of statistical tests in the Methods. In particular, please consider the reviewer's suggestion to use other statistical approaches such as the prevalence risk ratio instead of odds ratio.  These should be addressed in the revised manuscript.  

We look forward to receiving your revised manuscript.

Kind regards,

Peter R Thorne, CNZM PhD

Academic Editor

PLOS ONE

Journal Requirements:

3. Thank you for submitting the above manuscript to PLOS ONE. During our internal evaluation of the manuscript, we found significant text overlap between your submission and the following previously published works.

- http://www.audiologiabrasil.org.br/icaeia2010/ica2010_anais.pdf

- https://doi.org/10.3945/jn.110.122010

We would like to make you aware that copying extracts from previous publications, especially outside the methods section, word-for-word is unacceptable, even for works which you authored. In addition, the reproduction of text from published reports has implications for the copyright that may apply to the publications.

Please revise the manuscript to rephrase the duplicated text, cite your sources, and provide details as to how the current manuscript advances on previous work. Please note that further consideration is dependent on the submission of a manuscript that addresses these concerns about the overlap in text with published work.

Reviewers' comments:

Reviewer's Responses to Questions

**Comments to the Author**

1. Is the manuscript technically sound, and do the data support the conclusions?

Reviewer #1: Yes

Reviewer #2: Partly

2. Has the statistical analysis been performed appropriately and rigorously? 

Reviewer #1: Yes

Reviewer #2: No

3. Have the authors made all data underlying the findings in their manuscript fully available?

Reviewer #1: Yes

Reviewer #2: Yes

4. Is the manuscript presented in an intelligible fashion and written in standard English?

Reviewer #1: Yes

Reviewer #2: Yes

5. Review Comments to the Author

Reviewer #1: Thank you for your manuscript. Please find my comments and suggestions for your consideration below.

ABSTRACT:

Line 39: Participants reporting any past noise exposure at baseline (please add: were at a) greater risk of developing hearing loss 10 years later: multivariable-adjusted OR 1.67 (95% CI 1.19-2.33).

INTRODUCTION:

The term 'ageing' and 'aging' has been used interchangeably throughout this manuscript. Please use one spelling for consistency. My suggestion would be 'ageing'.

The acronym 'ARHL' is introduced on line 61. Please add in brackets when first mentioned on line 50, first sentence.

Line 75: please change 'expose' to 'exposure'.

RESULTS

(n=2015). 68 plus a further 15 (total 83) were excluded (line 151-153). Line 154 says that this resulted in a total of 1923 subjects but 2015 minus 83 = 1932. Can you clarify and account for the missing 10 please?

Line 155: '...679 participants who reported previous exposure to occupational noise and 1244 with no prior exposure'. This suggests that in total 679 participants reported occupational noise exposure. However, the 'N' for noise exposure in Tables 1 and 2 don't add up to 679. Please clarify.

Can you describe how you categorised the severity of noise exposure (tolerable and severe) in your methods after line 130. Was 'mostly quiet' treated as no exposure; 'tolerable but able to hear speech' as tolerable; and 'unable to hear anyone speaking' as severe?

DISCUSSION:

Line 253 to 255, '...and that this appears to be related in part to the difficulties imposed by such protection upon communication with other workers, especially in an emergency (REF?)' Please add reference.

Line 255: 'Indeed, in our study cohort we 'observed' only 10% of those exposed to occupational noise used any form of hearing protection devices'. I am not sure if you observed HPD use. Perhaps reword to: ' Indeed, in our study cohort 10% of those reporting exposure to occupational noise reported using any form of hearing protection devices'.

Line 269 to 272: Is almost a direct quotation from the Dobie paper. That was a 2008 paper. Do Audiologists currently not provide counselling and periodic audiometry for those exposed to occupational and non-occupational noise exposure?

Line 275 to 277: '...audiometric testing to measure hearing sensitivity rather than self-report...' Correct, but you relied on self-reported accounts of exposure to noise (duration, severity, etc). Could this not be a limitation of your study? You touch on it by comparing recall bias of people with and without hearing impairment.

My suggestion is to discuss (a paragraph) why hearing conservation programmes and efforts to prevent exposure to occupational noise has been successful or not successful. For example, line 263: 'Although, case studies show that significant reductions can be achieved, there is no evidence that this is realised in practice [27]'. Why?

REFERENCES:

There are a few dated references especially related to occupational noise and hearing protection use.. For example, reference 27. There is a recent Cochrane review that should be cited instead.

Reviewer #2: The paper is well written and the authors are clearly experts in this area. This article warrants publication but could benefit from some changes, namely in the statistical approach. My comments are meant to improve this piece.

Introduction

Line 50 – It seems something is missing, age-related hearing loss is more frequent compared to what?

The introduction’s content around the disconnect between findings of noise-induced hearing loss in animal models versus large observational trials is appreciated but it would benefit from further discussion as to why the authors believe this disconnect is present and why their data will improve the literature (i.e., is it a lack of long-term follow up, specifically?).

Methods

Lines 79-80 – it seems the dates of the study are incongruent – the first line says 1997-2004 but the next line indicates data collected in 1992-1994. Please clarify.

Was noise exposure measured only once?

The section on noise exposure would benefit from a concrete definition of how noise will be models (i.e., the specific variable definition) in the statistical models in addition to describing the questions asked.

For the authors’ consideration on the statistics: 1. Given the prevalence of hearing loss (i.e., hearing is not a rare event), it is likely that odds ratios overestimate the size effect, prevalence risk ratios would be more conservative and offer a more easily interpretable coefficient for the reader 2. Cox proportional hazard models could better examine the time-to-event incident hearing loss 3. Given the data available, linear fixed effects models could better characterize the longitudinal associations

Results/Discussion

These sections are great. The attempts to place the current findings within the context of other studies is important and the discussion of prevention is well done.

6. PLOS authors have the option to publish the peer review history of their article (what does this mean?). If published, this will include your full peer review and any attached files.

Reviewer #1: No

Reviewer #2: No

---

## [Author Response · Author response to Decision Letter 0]

11 Jun 2021

Reviewer #1:

ABSTRACT:

Line 39: Participants reporting any past noise exposure at baseline (please add: were at a) greater risk of developing hearing loss 10 years later: multivariable-adjusted OR 1.67 (95% CI 1.19-2.33).

Author response: As suggested by the Reviewer we have now reworded this sentence. 

INTRODUCTION:

The term 'ageing' and 'aging' has been used interchangeably throughout this manuscript. Please use one spelling for consistency. My suggestion would be 'ageing'.

Author response: As requested, we have now used the term ‘ageing’ throughout our manuscript.

The acronym 'ARHL' is introduced on line 61. Please add in brackets when first mentioned on line 50, first sentence.

Author response: Apologies, we have now removed this acronym and replaced it with ‘age-related hearing loss.’

Line 75: please change 'expose' to 'exposure'.

Author response: We have now changed it to ‘exposure’.

RESULTS

(n=2015). 68 plus a further 15 (total 83) were excluded (line 151-153). Line 154 says that this resulted in a total of 1923 subjects but 2015 minus 83 = 1932. Can you clarify and account for the missing 10 please?

Author response: Apologies, this was a typographical error it should have been ‘1932’ not ‘1923’ we have now corrected this in the Results section.

Line 155: '...679 participants who reported previous exposure to occupational noise and 1244 with no prior exposure'. This suggests that in total 679 participants reported occupational noise exposure. However, the 'N' for noise exposure in Tables 1 and 2 don't add up to 679. Please clarify.

Author response: The N (%) presented in tables 1 and 2 denote those who had any, mild or moderate-severe hearing loss according to the different noise exposure variables (duration and/or severity). However, not all 679 participants exposed to noise had a hearing loss, hence, the N presented in these tables do not add up to 679 (as we did not present data for the normal hearing function group).

Can you describe how you categorised the severity of noise exposure (tolerable and severe) in your methods after line 130. Was 'mostly quiet' treated as no exposure; 'tolerable but able to hear speech' as tolerable; and 'unable to hear anyone speaking' as severe? 

Author response: We now clarify in the Methods (lines 132-133) that: ‘Participants were classified as being exposed to ‘severe’ noise levels, if their response was that they were ‘unable to hear anyone speak’, all other responses were classified as a ‘tolerable’ level of noise.’

DISCUSSION:

Line 253 to 255, '...and that this appears to be related in part to the difficulties imposed by such protection upon communication with other workers, especially in an emergency (REF?)' Please add reference.

Author response: We have now added a reference for this statement. 

Line 255: 'Indeed, in our study cohort we 'observed' only 10% of those exposed to occupational noise used any form of hearing protection devices'. I am not sure if you observed HPD use. Perhaps reword to: ' Indeed, in our study cohort 10% of those reporting exposure to occupational noise reported using any form of hearing protection devices'.

Author response: Agreed. We have now reworded this sentence as suggested by the Reviewer.

Line 269 to 272: Is almost a direct quotation from the Dobie paper. That was a 2008 paper. Do Audiologists currently not provide counselling and periodic audiometry for those exposed to occupational and non-occupational noise exposure?

Author response: We have clarified this statement and have cited a 2018 reference to support this: ‘Moreover, audiologists and otolaryngologists see patients who have significant unprotected occupational and non-occupational noise exposure, and as such they play an important role in providing counselling, hearing protection and periodic audiometry for these patients [30].’

Line 275 to 277: '...audiometric testing to measure hearing sensitivity rather than self-report...' Correct, but you relied on self-reported accounts of exposure to noise (duration, severity, etc). Could this not be a limitation of your study? You touch on it by comparing recall bias of people with and without hearing impairment.

Author response: Agreed. We have now added the following study limitation (line 283): ‘Limitations of the study include potential systematic recall bias: participants self-reported duration and severity of noise exposure and participants…’ 

My suggestion is to discuss (a paragraph) why hearing conservation programmes and efforts to prevent exposure to occupational noise has been successful or not successful. For example, line 263: 'Although, case studies show that significant reductions can be achieved, there is no evidence that this is realised in practice [27]'. Why?

Author response: This is a good suggestion by the Reviewer and we have reworded this section in the Discussion (lines 266-276): ‘Currently, there is very low‐quality evidence that the use of hearing protection devices in well‐implemented hearing loss prevention programmes is linked to reduced hearing loss but this could not be demonstrated for other aspects, such as monitoring of noise levels, worker training or audiometry alone [28]. Additionally, engineering solutions such as new equipment, segregation of noisy equipment, installation of panels or curtains were shown to significantly reduce noise levels in control intervention case studies [28]. However, longitudinal research on the effects of engineering interventions to reduce noise is needed. For instance, field case studies with valid measures of personal noise doses of workers with prospective follow‐up would provide better evidence than what is currently available [28]. At a minimum, for an effective and successful hearing conservation program - noise surveys and monitoring, employee education, training, and motivation, hearing protection equipment, audiometric testing, and record-keeping, as well as noise control are important [29].’

REFERENCES:

There are a few dated references especially related to occupational noise and hearing protection use.. For example, reference 27. There is a recent Cochrane review that should be cited instead.

Author response: We have now cited the 2017 Cochrane Review reference rather than the 2009 review, additionally we have added some more recent references. 

Reviewer #2:

Introduction

Line 50 – It seems something is missing, age-related hearing loss is more frequent compared to what?

Author response: Agreed. We have now reworded this sentence as the following: ‘Age-related hearing loss is the most frequent communication disorder…’

The introduction’s content around the disconnect between findings of noise-induced hearing loss in animal models versus large observational trials is appreciated but it would benefit from further discussion as to why the authors believe this disconnect is present and why their data will improve the literature (i.e., is it a lack of long-term follow up, specifically?).

Author response: Yes, the reviewer is correct about the lack of long term data on the association between noise exposure and hearing loss. However, in lines 74-76 we have already discussed the reason for the disconnect and how our study improves the literature: ‘Despite the burden of noise and age-related hearing loss, there are very few large population-based studies that have evaluated both the cross-sectional and longitudinal contribution of noise exposure to age-related hearing loss. Therefore, we propose to explore the prevalence, 10-year incidence and progression of hearing loss associated with occupational noise exposure among older adults aged 50+ years at baseline.’ Moreover, we have expanded a sentence in the Introduction (lines 62-65) to also further clarify this disconnect: ‘There is a paucity of longitudinal population-based studies that have investigated noise exposure and hearing decline and observational studies have thus far provided equivocal findings on the relationship between noise exposure and subsequent hearing loss in adults.’

Methods

Lines 79-80 – it seems the dates of the study are incongruent – the first line says 1997-2004 but the next line indicates data collected in 1992-1994. Please clarify.

Author response: This is correct because in line 87-88 we indicate that while the Blue Mountains Eye Study started in 1992 (or BMES-1), hearing was only measured from BMES-2 onwards (i.e. from 1997-99). We have tried to clarify this better in the Methods now.

Was noise exposure measured only once?

Author response: No, noise exposure was measured throughout the 10-year period of the hearing study but for the purposes of the current report we analysed baseline noise exposure with the prevalence and incidence of hearing loss.

The section on noise exposure would benefit from a concrete definition of how noise will be models (i.e., the specific variable definition) in the statistical models in addition to describing the questions asked.

Author response: We have now clarified the specific variable definition (lines 129-132): ‘Based on whether a respondent answered ‘yes’ or ‘no’ to this question, they were classified as having any exposure to occupational noise or no exposure to occupational noise, respectively, and this was the specific variable included in statistical models.’

For the authors’ consideration on the statistics: 1. Given the prevalence of hearing loss (i.e., hearing is not a rare event), it is likely that odds ratios overestimate the size effect, prevalence risk ratios would be more conservative and offer a more easily interpretable coefficient for the reader 2. Cox proportional hazard models could better examine the time-to-event incident hearing loss 3. Given the data available, linear fixed effects models could better characterize the longitudinal associations

Author response: We have taken these suggestions by the Reviewer into consideration, and we now present risk ratios obtained by using linear fixed effects for longitudinal data (i.e. using Poisson Regression approach to prospective studies with Log as link function) in Table 3 and all text in the manuscript has been updated accordingly.

---

## [Editor Report · Decision Letter 1]

15 Jul 2021

Workplace Noise Exposure and the Prevalence and 10-Year Incidence of Age-Related Hearing Loss

PONE-D-21-00280R1

Dear Dr. Gopinath,

We’re pleased to inform you that your manuscript has been judged scientifically suitable for publication and will be formally accepted for publication once it meets all outstanding technical requirements.

Kind regards,

Peter Rowland Thorne, CNZM PhD

Academic Editor

PLOS ONE

Additional Editor Comments (optional):

The reviewers comments have been amended appropriately.

There are a few minor typographical errors as listed which should be addressed during processing of the manuscript:

Line 52 Ageing-related degenerati(on) may..

Line 232 ..the cochlea(r).

Line 266 program - .Check consistency of spelling throughout (program vs programme
---

## [Editor Report · Acceptance letter]

22 Jul 2021

PONE-D-21-00280R1 

Workplace Noise Exposure and the Prevalence and 10-Year Incidence of Age-Related Hearing Loss 

Dear Dr. Gopinath:

I'm pleased to inform you that your manuscript has been deemed suitable for publication in PLOS ONE. Congratulations! Your manuscript is now with our production department. 

Kind regards, 

on behalf of

Dr. Peter Rowland Thorne 

Academic Editor

PLOS ONE